# OpenReview forum: "Breaking the Mirror: Activation-Based Mitigation of Self-Preference in LLM Evaluators"
_NeurIPS.cc/2025/Workshop/Reliable_ML — NeurIPS 2025 - Reliable ML Workshop_

### Official Review · Reviewer_WbH9 · 2025-09-17
**Overall positive evaluation**

**Rating:** 6
**Confidence:** 3

**Review:**

Summary
The paper presents a lightweight, activation-based safeguard mechanism designed to address the problem of self-preference bias in large language models (LLMs) when these models are used in automated evaluation settings. The authors argue that when LLMs are asked to evaluate outputs, they often display a tendency to prefer their own generations over those produced by alternative systems. To mitigate this issue, the authors introduce a curated dataset specifically tailored to highlight instances of self-preference bias. They then use this dataset to benchmark their safeguard method against two commonly referenced baselines: prompting strategies and Direct Preference Optimization (DPO). According to their empirical findings, the proposed safeguard is able to substantially reduce instances of self-preference bias, achieving deterrence in up to 97% of evaluated cases. At the same time, however, the authors acknowledge a drawback: the method sometimes over-corrects, leading to legitimate evaluations being flipped erroneously, which may limit reliability in certain practical applications.

Strengths
One of the most notable strengths of the work lies in its novelty. To the best of my knowledge, prior research has not proposed such an activation-based safeguard for addressing bias in automated LLM evaluation, making this contribution both timely and original. The empirical study is also conducted with care: the methodology is systematic, the analysis is thorough, and the experiments are presented in a transparent manner that avoids overstating claims. The curated dataset itself is an additional valuable contribution to the community, as it enables reproducibility and future research on this underexplored issue. Furthermore, the work is clearly relevant for ongoing discussions around the reliability of evaluation methods in scenarios where data and human labels are imperfect. By focusing on mitigating self-preference bias, the authors make progress toward ensuring that automated evaluations are more trustworthy and less skewed by the limitations of the underlying model.

Weaknesses and Limitations
Despite the positive aspects, there are some limitations worth highlighting. First, the comparison to optimization-based methods raises concerns about efficiency. In particular, methods like DPO appear to require significantly fewer training samples while still achieving performance that is broadly comparable to the safeguard proposed in this paper. This difference in sample efficiency may make it difficult to justify the activation-based approach in situations where data or compute resources are limited. Second, the applicability of the proposed method seems restricted primarily to binary evaluation settings. While this is a useful test case, it leaves open questions about how well the approach would generalize to more complex evaluation tasks, such as multi-class judgments or ranking-based assessments, which are common in realistic evaluation pipelines. Without further exploration of these broader contexts, the practical utility of the method remains somewhat constrained.

Suggestions for the Authors
The paper would be strengthened by expanding on several of the points that are already briefly mentioned in the Discussion and Future Work section. For example, it would be useful to see a deeper investigation into the trade-off between bias reduction and erroneous flipping of legitimate evaluations, perhaps through a more fine-grained error analysis. Similarly, extending the method beyond binary evaluation tasks to a wider variety of evaluation formats could broaden the impact of the work. Finally, given that the safeguard requires more samples compared to optimization-based methods, exploring ways to improve its efficiency—such as hybrid approaches or more targeted sampling strategies—could address some of the concerns raised in the limitations.

Ethics
No significant ethical concerns are apparent in this work. The authors are transparent about the limitations of their method, and the focus of the paper is on improving fairness and reliability in automated evaluation.

---

### Official Review · Reviewer_DJpu · 2025-09-18
**Activation-Based Mitigation of Self-Preference: Promising but Limited**

**Rating:** 5
**Confidence:** 3

**Review:**

## Brief summary:
This paper investigates the issue of self-preference in LLM-based evaluators and applies two activation-steering methods to mitigate it. Their experiments show that these approaches can correct a large proportion of biased judgments, achieving high flip rates for unjustified self-preference. However, the methods also exhibit instability, occasionally overturning correct evaluations and introducing unintended errors.

## Strengths. Novelty, rigor, empirical/theoretical quality, clarity, relevance to reliability with imperfect data
1. The proposed correction method is lightweight and can be applied at inference time without retraining, significantly reducing deployment overhead.
2. Compared to baseline approaches on the XSUM dataset, the method demonstrates strong correction performance.

## Weaknesses / Limitations. Missing comparisons/ablations, unclear assumptions, proof gaps, failure modes, scope limits.
1. The experiments are limited to the XSUM dataset. To demonstrate broader applicability, evaluations on additional benchmarks are necessary.
2. While the proposed method successfully corrects certain biased judgments, it also introduces new errors by overturning correct decisions.
3. The evaluation primarily focuses on correction rate as the main metric. However, beyond the risk of introducing new errors, the study does not examine whether these corrections translate into improvements in downstream tasks—an aspect that deserves greater attention.

## Suggestions for Authors. Specific things that would improve the paper:
1. extensive experiment on different dataset is needed
2. it will be better to prevent the method from incorrectly affect the judgment

## Ethics (if applicable). Note any concerns (about privacy, fairness, misuse, sensitive data use) and suggested mitigations.
N/A